# Scalable dissolution-dynamic nuclear polarization with rapid transfer of a polarized solid

Karel Kouřil [1], Hana Kouřilová[1], Samuel Bartram[1], Malcolm H. Levitt [1] & Benno Meier [1]

In dissolution-dynamic nuclear polarization, nuclear spins are hyperpolarized at cryogenic temperatures using radicals and microwave irradiation. The hyperpolarized solid is dissolved with hot solvent and the solution is transferred to a secondary magnet where strongly enhanced magnetic resonance signals are observed. Here we present a method for transferring the hyperpolarized solid. A bullet containing the frozen, hyperpolarized sample is ejected using pressurized helium gas, and shot into a receiving structure in the secondary magnet, where the bullet is retained and the polarized solid is dissolved rapidly. The transfer takes approximately 70 ms. A solenoid, wound along the entire transfer path ensures adiabatic transfer and limits radical-induced low-field relaxation. The method is fast and scalable towards small volumes suitable for high-resolution nuclear magnetic resonance spectroscopy while maintaining high concentrations of the target molecule. Polarization levels of approximately 30% have been observed for 1-$^{13}$C-labelled pyruvic acid in solution.

[1] School of Chemistry, University of Southampton, Southampton SO17 1BJ, United Kingdom. Correspondence and requests for materials should be addressed to K.K. (email: k.kouril@soton.ac.uk) or to B.M. (email: b.meier@soton.ac.uk)

Nuclear magnetic resonance (NMR) is a powerful, non-invasive analytical technique. It is broadly divided into two branches—NMR spectroscopy and magnetic resonance imaging (MRI). NMR spectroscopy provides structural and dynamic information on objects ranging from small molecules to proteins, while MRI provides spatial information with sub-millimeter resolution and has become indispensable for diagnostic medicine. The sensitivity of NMR however is very low[1], in particular because the interaction between nuclear spins and the applied magnetic field is orders of magnitude smaller than the thermal energy available at room temperature. Consequently, the nuclear spins are only weakly polarized. In thermal equilibrium at room temperature only 1 in 100,000 proton spins contributes to the signal that is recorded in an MRI scanner at 3 Tesla. Other nuclei exhibit a smaller gyromagnetic ratio, and hence an even smaller polarization. For example the polarization of the $^{13}$C isotope of carbon at a given magnetic field is four times smaller than that of protons.

Dissolution-dynamic nuclear polarization (D-DNP), first described by Ardenkjær-Larsen and co-workers in 2003[2], can increase the sensitivity of NMR by orders of magnitude. In D-DNP, a sample containing the molecule of interest is mixed with a suitable radical and polarized at a field of several Tesla and a temperature of ~1 K by transferring the substantially larger electron spin polarization to the nuclei using microwave irradiation. Subsequently, a jet of hot solvent, propelled by pressurized He gas, is injected into the cryostat, dissolving the hyperpolarized substance. The hyperpolarized solution is flushed through a narrow transfer tube to the site of observation (an NMR or MRI instrument). The attainable signal enhancements of 10,000 and more have enabled a tracking of in vivo metabolism in humans[3–5]. Hyperpolarized metabolites have potential uses in the monitoring of treatment response at early stages of cancer therapy[5–7].

Although notable applications of D-DNP to biomolecules exist[8–11], D-DNP has not become a tool used routinely in NMR spectroscopy. In the past major limitations of D-DNP have been long polarization times, low polarization levels, and high operational costs for helium and staff. New concepts such as cross-polarization and gated microwave frequency sweeps reduce the polarization time, and yield polarization levels approaching unity[12,13] in less than 30 min. Cryogen-free and more automated systems[14–16] further increase the performance of D-DNP.

Two substantial limitations remain associated with the way the hyperpolarized sample is dissolved and transferred.

Firstly, solvent volumes of 3–5 mL or more are needed to prevent freezing in the cold regions of the polarizer, leading to a typically 100-fold dilution of the target molecule and a corresponding reduction in signal strength. Substantially higher concentrations have been achieved by adding an immiscible phase to the dissolution medium, but, when using standard 5-mm NMR tubes, solvent separation is not complete, and residuals of the immiscible phase lead to unacceptable line broadening of hundreds of Hz[17]. The immiscible phase is toxic to many molecules of interest, further limiting the scope of such dual-solvent approaches.

Secondly, the transfer of a liquid through a thin tube is intrinsically slow. Most instruments require up to 10 s for sample transfer, and high-speed transfer systems employing pressurized HPLC loops still require 1–4 s, often at the expense of even greater sample dilution[18,19]. This is a substantial time compared to the spin-lattice relaxation time $T_1$ with which the hyperpolarization decays, especially for larger molecules that exhibit shorter $T_1$ values.

An alternative approach is to transfer the sample in the solid state. Hirsch et al. have shown that it is possible to brute-force polarize $^1$H nuclei at high magnetic field and low temperatures, and transfer the proton spin polarization to $^{13}$C nuclei during a 1 s sample passage through low-field, followed by sample dissolution near the target magnet[20,21]. However, the $^{13}$C polarization that can be attained in this way was only 0.15%.

DNP can yield near-unity spin polarization, but requires radicals as a source of polarization. Radicals cause extremely rapid relaxation at low fields and elevated sample temperatures. Samples for brute-force hyperpolarization are therefore deoxygenated to remove any paramagnetic impurities and kept under inert atmosphere[20,21].

Here we show that spin polarization in excess of 30% can be retained if a sample containing radicals is transferred very rapidly in the solid state, and dissolved only at the point of use. This scheme, named bullet-DNP, overcomes the current requirement to dissolve hyperpolarized samples in the polarizer, which has been recognized as a limitation of dissolution-DNP[22]. The two key advantages of extracting the sample in its solid form are scalability and speed. Firstly, in conventional D-DNP the cryogenic sample environment rapidly cools the injected solvent, leading to blockages if solvent amounts of less than 3 mL are used. Since NMR spectroscopy requires only 700 μL or less, this leads to substantial sample dilution, as well as smaller signals and/or higher costs for the substrates under investigation. In bullet-DNP only small solvent amounts are required since the solvent is never exposed to the cryogenic environment inside the polarizer. Here we use solvent amounts of 700 μL to comfortably fill 5 mm NMR tubes. Secondly, the transfer of a solid through a tube is very fast, even at moderate drive pressures.

## Results

**Apparatus**. The apparatus for bullet-DNP is shown in Fig. 1.

While the polarizer shares many elements with previously reported systems[2,23], the two distinguishing features are a provision to rapidly eject samples using pressurized gas and a carefully engineered magnetic tunnel along the entire transfer path. A rendering of the lower part of the DNP insert is shown in panel (**b**) of Fig. 1. The DNP insert comprises three steel tubes for helium drive gas, sample insertion and ejection, and microwave irradiation. The drive and sample tubes are 1/4" steel tubes with 4.2 mm ID, while the tube carrying the microwave is a 3/8" tube. The bottom of the DNP insert comprises three brass pieces. The bottom brass piece connects the drive tube with the sample tube via an internal channel that is connected to the outside helium bath during normal operation but closes during ejection of the sample using an excess flow valve. Thus during normal operation liquid helium flows into the sample tube for efficient sample cooling. The center brass part carries a gold-plated mirror for the microwave irradiation and supports a printed circuit board for tuning and matching the radio-frequency (RF) saddle coil. The RF coil is wound onto a PEEK coil support that hosts the bullets. The bullets are miniature cylindrical buckets made from PTFE. The bullet is 12 mm tall, with an outer diameter of 3.9 mm and a 10 mm deep, 3.5 mm diameter concentrical bore to accommodate the sample. Each bullet can accommodate sample volumes up to 80 μL. The PEEK coil support is transparent for the microwave and RF fields required to polarize the sample and monitor the polarization level, but confines the helium gas inside the transfer path during sample ejection. Brass screws push the center brass piece onto the PEEK coil support that is thereby pushed onto the bottom brass piece. The sample tube fits into the top of the coil support that compresses on cooling, affording a good seal of the transfer path.

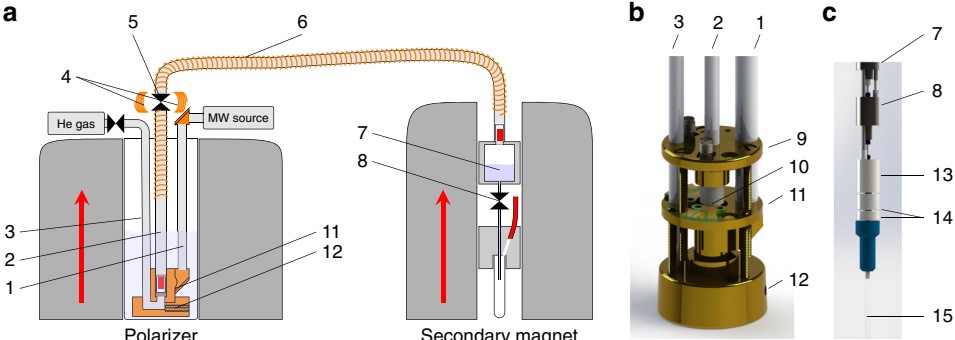

**Fig. 1** Bullet-DNP equipment. **a** Sketch of the overall setup, comprising a polarizer magnet with cryostat and DNP insert, a transfer tube (6) with a solenoid along its entire length, and an injection device inside a liquid-state NMR magnet. The polarity of the superconducting magnets is indicated by the red arrows. The direction of the applied pulsed magnetic field along the transfer changes at the valve (5), where two orthogonal pairs of Helmholtz coils (4) are used to rotate the field. **b** Rendering of the lower part of the DNP insert showing the drive gas tube (3) on the left, the main sample tube (center, 2) and the microwave tube (1). The central brass piece (11) hosts small circuitboards for tuning and matching. SMA connectors are soldered to the top brass piece (9) that also pushes the coil support (10) onto the bottom brass piece (12) to seal the transfer path. **c** Rendering of the lower part of the injection device with a mounted NMR sample tube. The bottom of the solvent reservoir (7) can be seen at the top of the image, followed by the pinch valve (8) and a 3D printed nylon piece (13) with an internal channel that connects a 3/16″ steel tube to the NMR sample tube to facilitate evacuating the latter. Two further nylon pieces (14) are screwed onto the NMR tube (15) using a compression fitting, and facilitate a fast exchange of NMR tubes in between experiments

A 1/4″ inch T-union (Swagelok, UK) on top of the plug valve (SS-4P4T, Swagelok) at the top of the sample tube is used to apply a small stream of helium gas when loading the bullet into the cold sample space, thereby preventing contamination of the system with air. After loading the sample, the flexible transfer tunnel is connected at the top of the Swagelok T-union.

In the experiment presented here, the sample is shot through a 3.2 m long transfer tunnel that is connected to the top of the DNP insert and extends 50 cm into the injection device in the secondary magnet. A minimum bend radius of 0.5 m is chosen to limit sample deceleration in the tunnel bends. A solenoid wound along the entire transfer path is energized with a current of 60 A prior to sample ejection, using a 15 kW power supply (Keysight Technologies, US). A solenoid is also wound around the sample tube inside the polarizer, ensuring that the field experienced by the sample never drops below 75 mT. The solenoids are made by tightly winding 1 mm diameter insulated copper wire (RS) onto the transfer tube, leading to an estimated field of 75 mT for a current of 60 A. The polarizer and the secondary magnet in our lab both have the same polarity. Therefore, a reversal of the field direction along the transfer path is required. Any change in field direction should occur adiabatically, meaning that the Larmor frequency $\omega_L$ should always be substantially larger than the frequency of the rotation of the field as seen from a frame fixed with respect to the moving bullet[24]. In our setup, this is achieved using two pairs of Helmholtz coils around the plug valve at the top of the sample tube. The individual coils have a diameter of ~6 cm. The first pair of Helmholtz coils is used to apply a transverse field, the second pair is a longitudinal anti-Helmholtz pair. Inside the polarizer the magnetic field is applied along the direction of travel. Following passage through the Helmholtz coils the magnetic field is applied opposite to the direction of travel. The field rotation is carried out over a distance of ~6 cm (see Supplementary Fig. 1 for a schematic drawing of the Helmholtz coils). At a bullet speed of 100 m/s this corresponds to a frequency of ~1 kHz, well below the Larmor frequency $\omega_L/(2\pi) = 800$ kHz of $^{13}$C at 75 mT. Note that the applied field of 75 mT also ensures that the Zeeman interaction dominates other spin interactions—the spins are quantized along the applied field at all times. In particular, the field is also sufficiently strong to substantially suppress thermal mixing of the proton and carbon spins in the sample[20]. This is

important, since in singly labeled $^{13}$C pyruvic acid the $^1$H spin bath has a ~50 times larger heat capacity than the $^{13}$C bath[20]. Thermal contact would therefore rapidly quench the $^{13}$C hyperpolarization.

The passage of the bullet through the transfer tube is monitored using optical barriers[19] at the entrance and exit of the transfer tunnel. For the experiment described here, the drive gas pressure was set to 10 bar, leading to a transfer time of ~70 ms. Preliminary experiments show that the transfer time may be reduced further by increasing the drive gas pressure. The sample is only ejected after the transfer coils are fully energized. The time dependence of the current in the coils, and the signals from the optical detectors are shown in Fig. 2a, b, respectively. The optical sensors are located ~40 cm above the field centers of the two main magnets.

Two different injection devices have been constructed. A simple device has been used for observations of quantum-rotor-induced polarization[25,26]. It consists of a 3D printed receiver that connects to the NMR tube at the bottom, and to the transfer tube at the top. Inside the receiver a recession in diameter retains the bullet - the sample itself carries sufficient momentum to slide out of the bullet and into the NMR tube that is preloaded with solvent. A similar device has been described by Hirsch et al.[21]. Venting slots at the top of the receiver provide an escape path for the pressurized helium gas. This device is fast and works well for organic solvents, but the impact of the sample on aqueous solutions often causes bubbles and leads to inhomogeneous solutions, in particular when using 5 mm NMR tubes.

We, therefore, constructed a new injection device, shown in Fig. 1c. In this device the bullet is also retained, but the sample itself is shot into a solvent reservoir located above the NMR tube. The solvent reservoir is machined from titanium and is equipped with a heater and a temperature sensor. With an internal diameter of 10 mm it facilitates fast dissolution and mixing[10]. A 1/8″ OD PTFE tube connects the solvent reservoir to a home-built valve in which a piece of silicon tubing is compressed with pressurized gas to block flow. The output of this valve is connected to an evacuated 5 mm NMR tube using a further piece of PTFE tubing. The valve opens upon releasing the pressure, and the solution is sucked from the solvent reservoir into the NMR tube. The evacuating line is closed to prevent boiling of the solution at low pressure. The filling of the NMR tube with

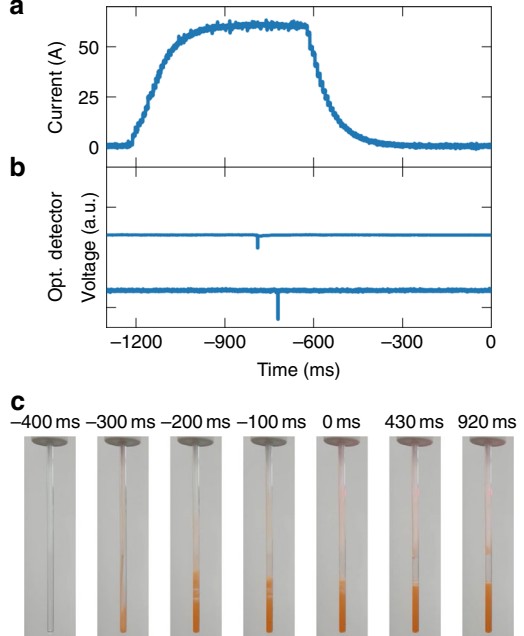

**Fig. 2** Timing of the sample ejection, relative to the start of the NMR data acquisition at $t = 0$. At $t = -1200$ ms the coil along the transfer path is energized. The voltage across the coil is ramped up to 200 V in ~150 ms. The current (panel **a**) takes ~200 ms to reach its maximum, and subsequently decreases slightly as the coils resistance increases due to resistive heating. At $t = -900$ ms the sample ejection routine is executed. The actuator chain comprises a mechanical relay, a pilot valve and an actuated plug valve, leading to a delay of another 100 ms before the bullet passes the first optical detector (panel **b**, top curve) near its start position, at $t = -800$ ms. ~70 ms later the bullet passes the second optical detector inside the second magnet (panel **b**, bottom curve), where the sample is dissolved in pre-heated solvent. At $t = -500$ ms the pinch valve in the injector is opened, and the sample is sucked into an evacuated NMR tube. To demonstrate the loading of the NMR tube, a bullet containing 60 μL of a 1:1 mixture of red dye and pyruvic acid was shot from the polarizer into the injection device that was preloaded with 700 μL of degassed $H_2O$, heated to 50 °C. Selected frames from a video taken as the solution flows into the NMR tube are shown in panel **c**. The time origin is the same as in panels **a**, **b**. The NMR acquisition is triggered at $t = 0$

aqueous solution is shown in Fig. 2c. The filling of the tube is essentially complete at $t = 0$ ms, ~860 ms after the bullet passes the optical barrier at the tunnel entrance inside the polarizer, and NMR acquisition is triggered. Remaining bubbles at the top of the sample collapse within the first second after the trigger. After the experiment, the injection device is removed from the magnet, the empty bullet is removed manually and the NMR tube is exchanged for a new one.

**Solid-state polarization**. The solid-state polarization in our polarizer is estimated as $50 \pm 10\%$, based on a comparison of the hyperpolarized signal to the thermal signal measured at a temperature of 4.3 K. This polarization level is in the expected range for the polarization of neat pyruvic acid at 6.7 T and 1.4 K[14,27].

**Liquid-state polarization**. A $^{13}$C NMR spectrum of 80 μL hyperpolarized 1-$^{13}$C-labeled pyruvic acid after rapid transfer and dissolution in 700 μL of degassed $D_2O$ is shown in Fig. 3. The spectrum exhibits two strongly enhanced peaks that are due to the presence of pyruvic acid in both the oxo and the hydrated form[28]. As described previously[2], substantial polarization levels

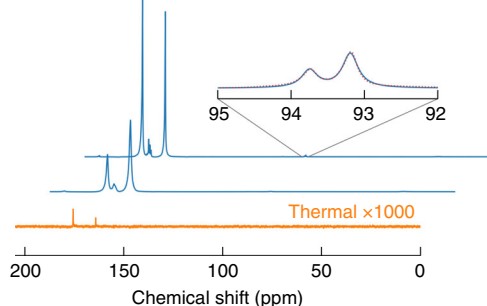

**Fig. 3** Two $^{13}$C NMR spectra of hyperpolarized 1-$^{13}$C-pyruvic acid recorded immediately, and two seconds after rapid transfer, dissolution in degassed $D_2O$ and injection (lower and upper blue curve, respectively). The spectra were recorded with a flip angle of ~10°. The spectra show two strongly polarized peaks at 175.7 and 164.2 ppm due to the presence of pyruvate and pyruvate hydrate, respectively. The inset is a close-up of the signal at ~93.5 ppm which is due to the natural abundance carbonyl of pyruvate hydrate. This signal is split into a doublet by the 67 Hz J-coupling to the (labeled) carboxyl carbon. The asymmetry in the doublet is modeled with two Lorentzians (red dotted line) and yields an estimate for the $^{13}$C polarization of $32 \pm 1\%$. The bottom, orange line shows a 1000-fold magnification of the thermal equilibrium signal (4 transients averaged)

lead to an asymmetry in the doublet of nearby J-coupled spins. The inset of Fig. 3 shows a close-up of the doublet at 93.5 ppm, which is due to the natural abundance carbonyl carbon of pyruvate hydrate. The asymmetry of the doublet yields an estimate for the $^{13}$C polarization of $32 \pm 1\%$[2]. A direct comparison of the hyperpolarized signal with the thermal equilibrium signal yields a slightly smaller estimate for the $^{13}$C polarization of 28%.

The spectrum also shows a set of strongly enhanced peaks between the two main peaks at around 170 ppm. These signals are attributed to impurities such as para-pyruvate and zymonic acid[28–30].

At present the violent impact of the solid sample onto the solvent leads to the incorporation of fine helium bubbles that lead to line broadening of ~30 Hz 2 s after dissolution. Schemes to rapidly remove helium bubbles during injection into the NMR tube are currently being explored.

**Sample heating during transfer**. To estimate the sample heating during the transfer, a platinum temperature sensor was inserted into a bullet containing pyruvic acid. The bullet containing the sensor and the pyruvic acid was frozen in liquid nitrogen and lowered into a helium dewar, where it equilibrated at ~20 K. Then, it was rapidly brought to the top of the dewar where it was exposed to rapidly flowing ambient temperature helium gas. This experiment, detailed in Supplementary Methods and Supplementary Fig. 2, shows that the sample heats up rapidly at low temperatures, with a typical heating rate of 60 K/s at 30 K.

## Discussion
We have shown that it is possible to transfer solid hyperpolarized samples containing a high concentration of radicals while retaining a substantial fraction of the nuclear spin polarization. The sample transfer is carried out in less than 100 ms, which limits sample heating to temperatures below 30 K. A pulsed coil arrangement ensures an adiabatic transfer, prevents thermal mixing and limits low-field relaxation. The amount of solvent can be adjusted to match volumes required for high-resolution NMR spectroscopy.

Even without further optimization the approach presented here is attractive for a range of applications, in particular NMR spectroscopy, and imaging of small animals, where the amount of solution that can be used is often limited to 0.5 mL or less[31]. For

pre-clinical applications the sample could be dissolved either inside an injection unit in the MRI scanner, or in a nearby Hallbach magnet. With fully automated sample ejection we have found bullet-DNP to be robust, and when limiting the polarization time at 1.4 K to 30 min, we have been able to perform up to 10 successful experiments in a single day.

State-of-the-art conventional D-DNP currently achieves up to 70% polarization in solution, when applied to molecules with long $T_1$ such as pyruvate[16], while bullet-DNP currently achieves ~30% polarization in solution. The lower polarization is due to a lower initial solid-state polarization of ~50%, and losses of polarization during transfer and dissolution.

Numerous options exist to both increase the polarization in the solid state and reduce polarization losses further. Proven strategies to increase the polarization level in the solid are the use of cross-polarization[12], frequency sweeps[16], or lower temperatures[32].

Strategies to reduce polarization losses during transfer include the (partial) use of permanent magnets to afford stronger transfer fields[33], and/or a cooling of the transfer path[21]. Polarization losses due to the presence of radicals may be reduced if the sample is divided into radical free parts and radical rich parts, although this approach has to date led to smaller polarization levels in solution[22]. Polarization losses may also be reduced to an insignificant level if UV radicals are used to polarize the material[34]. Preliminary data indicate that the polarization is best preserved at the highest transfer fields. The dependence of polarization transfer efficiency (defined as the ratio of polarization in solution and in the solid state) on transfer speed is more difficult to assert since different drive pressures may also affect the sample temperature, and thereby the relaxation during the transfer.

Bullet-DNP is scalable towards almost arbitrarily small samples and hence we expect it to yield outstanding mass sensitivity when combined with miniaturized NMR detectors[35,36].

Bullet-DNP may also have advantages when polarizing larger samples, e.g., for clinical studies. The transfer in the solid enables a combination of DNP with low-field thermal mixing (LFTM)[20], where polarization is transferred from protons to low-$\gamma$-nuclei simply by shuttling the sample through a region of low field. Such a strategy does not suffer from the sample size limitations that exist for cross-polarization[12]. While a modified design for larger substrates would be required for clinical studies, the use of LFTM may lead to more rapidly, more strongly polarized substrates for applications spanning from spectroscopy to clinical research.

## Methods

**Sample preparation**. A 20 mM solution of OX063 trityl radical (Oxford Instruments, UK) in neat 1-$^{13}$C pyruvic acid (Sigma Aldrich, US) was prepared. A volume of 80 μL of this solution was pipetted into a bullet that is then immersed in liquid nitrogen to freeze the solution.

**Sample loading and polarization**. The bullet was loaded into the sample tube of the DNP insert, and the sample space was filled with liquid helium. Subsequently, the needle valve on the helium transfer line was closed and the temperature decreased to typically 1.4 K, with a hold-time of more than four hours. The sample was polarized using microwave irradiation at 187.880 GHz, and the polarization buildup was monitored using a coil that is approximately tuned and matched to the $^{13}$C resonance frequency of 71.8 MHz.

**Polarizer magnet and cryostat**. Our DNP insert is mounted in an Oxford Instruments Spectrostat flow cryostat that is placed into a 6.7 Tesla magnet (Bruker). A KF50 pumping line connects the DNP insert to a 250 m³/h Roots pump, enabling temperatures down to 1.4 K.

## Data availability

The raw data used to generate the figures in this manuscript and in the supplementary material are available from University of Southampton e-Prints with the identifier https://doi.org/10.5258/SOTON/D0837. All other data that support the findings of this study are available from the corresponding author upon reasonable request.

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

## Acknowledgements

We would like to thank John Owers-Bradley and James Kempf for discussions. This research was supported by EPSRC (UK), grant codes EP/M001962/1, EP/P009980, EP/R031959/1, and the Wolfson Foundation.

## Author contributions

B.M. conceived the idea. B.M. and K.K. designed and built the apparatus, with contributions by H.K. K.K., H.K., S.B. and B.M. performed experiments and analyzed data. K.K., H.K., and B.M. wrote the paper. M.H.L. provided advice and co-wrote the paper.

## Additional information

**Competing interests:** B.M., K.K., and H.K. are co-founders of HyperSpin Scientific Ltd, UK. The remaining authors declare no competing interests.

