## [Peer Review File · Nature Communications]

Reviewers' comments:

Reviewer #1 (Remarks to the Author):

In this paper, the authors demonstrate a new method for dissolution DNP (D-DNP).

In conventional D-DNP, the hyperpolarized solid is dissolved in a magnet for DNP and after transferred into a magnet for high resolution NMR spectroscopy or MRI.

There are two issues on the sample size and the transfer speed.

The new method addresses these issues.

The technological breakthrough to resolve these issues is of significant importance in DNP and NMR spectroscopy and opens up new scientific fields in chemistry, biochemistry and biomedicine.

However, the present version of the paper is not convincing and suitable for publication in Nature Communications.

(1) Originality.

In the new method, the hyperpolarized solid is transferred into a magnet for high resolution NMR and after dissolved. The idea has already demonstrated in [22]. The differences are using brute-force hyperpolarization and the transfer speed. First, the authors should cite the paper earlier (e.g. at the beginning of 4th paragraph) and clarify the originality of [22]. The readers of [22] imagine that a higher speed is achieved by more expensive pneumatic technology. Imaginable and continuous development is not suitable for Nature Communications. The authors should show issues of the previous method and must convince that their paper has include a discontinuous development. To show result which can be obtained only by the instrumentation may be very good. Key point may be relaxation. The relaxation budget should be more quantitatively discussed. What is the relaxation time of pyruvate in liquid and solid in 75 mT? What is the temperature change in the transfer tube? How is relaxation in liquid? Where is second optical detector?

(2) 'Scalable' in title.

The audience except for specialists do not understand the meaning and significance of 'scalable' before read introduction. They may think that larger is usually more difficult. If the authors would like to use it, they should explain why small sample is needed and difficult.

(3) Methods.

Too technical for Nature Communications. The audience would like to pick-up key points of the work, which is prevented by the technical sentence which does not affect reproducibility. The authors should move the technical sentence to supplementary information. E.g. RS components and Python, etc.

Also, the explanation of the instrumentation is difficult to understand. To number 1, 2, and 3 (or A, B, and C) for each important element and cite these in the sentence makes the explanation easier to understand.

Reviewer #2 (Remarks to the Author):

This paper describes development and demonstrations of dissolution NMR based on a game-changing idea. That is, samples in the solid-state with spin-hyperpolarized nuclei is shot from the region where nuclear hyperpolarization is performed into another place where enhanced NMR signal is measured. Unlike the previous strategy in which the frozen sample is rapidly dissolved with hot solvent before being transported to the NMR-measurement section, the authors firstly transfer the solid sample, then dissolve it, and finally measure NMR in the liquid state. To maintain the nuclear-spin polarization, the transfer is made as rapid as that of bullet-shooting, and in addition, the sample is led through a tunnel of a switchable magnetic field. The advantages of reversing the order of operations (i.e., transfer followed by dissolution instead of dissolution followed by transfer) are appropriately discussed, and preliminary experimental demonstrations are presented.

The idea behind this work is rather straightforward. Nevertheless, the activity of putting it into practice is very aggressive, and I find the work beautiful and praiseworthy. Even though what are already written in the manuscript are clear, detailed descriptions, in particular about the following important issues, are missing:

- It is difficult for the reader to get Fig. 1(b)(c). The authors should indicate clearly the individual components that are important in Fig. 1(b)(c).
- Discussions on the minimum curvature of the tunnel for bullet-transfer are necessary.

- The Helmholtz coil for field reversal, which plays an important role, is explained only briefly in the main text, and it is very difficult to imagine how it works. A visual (photo and/or drawing) explanation is highly desirable.

- The asymmetric feature of the J-splitting resonance line (Fig. 3), which gives important information as to the attained spin polarization, would be obtained only when the observing spin packets are selectively excited, or the tip angle of the excitation pulse is much smaller compared to 90 degree. However, no comment is found.

Overall, I believe the topic of the paper and the authors' achievement are impressive and suitable for publication in Nat. Comm., whereas the presentation needs revision. A separate supplementary material describing technical details would be helpful.

Response To Reviewer's Comments

We would like to thank the reviewers for their comments. Reviewer 2 is very positive, but reviewer 1 raises a question regarding the novelty of our approach. A detailed response to the individual comments is included below.

1 Reviewer 1

Reviewer: In this paper, the authors demonstrate a new method for dissolution DNP (D-DNP). In conventional D-DNP, the hyperpolarized solid is dissolved in a magnet for DNP and after transferred into a magnet for high resolution NMR spectroscopy or MRI. There are two issues on the sample size and the transfer speed. The new method addresses these issues.

The technological breakthrough to resolve these issues is of significant importance in DNP and NMR spectroscopy and opens up new scientific fields in chemistry, biochemistry and biomedicine. However, the present version of the paper is not convincing and suitable for publication in Nature Communications.

(1) Originality. In the new method, the hyperpolarized solid is transferred into a magnet for high resolution NMR and after dissolved. The idea has already demonstrated in [22]. The differences are using brute-force hyperpolarization and the transfer speed. First, the authors should cite the paper earlier (e.g. at the beginning of 4th paragraph) and clarify the originality of [22]. The readers of [22] imagine that a higher speed is achieved by more expensive pneumatic technology. Imaginable and continuous development is not suitable for Nature Communications. The authors should show issues of the previous method and must convince that their paper has include a discontinuous development. To show result which can be obtained only by the instrumentation may be very good. Key point may be relaxation. The relaxation budget should be more quantitatively discussed. What is the relaxation time of pyruvate in liquid and solid in 75 mT? What is the temperature change in the transfer tube? How is relaxation in liquid? Where is second optical detector?

Response: It is correct that Hirsch et al. have shown that polarization can be transferred in the solid state. However, as Hirsch et al. describe, the presence of even small quantities of radicals lead to very fast relaxation in their experiment. The observation by Hirsch et al. would therefore seem to rule out the possibility of combining DNP (where substantial quantities of radicals are required by definition) with a solid transfer. The experiment by Hirsch et al. relies on thermal mixing, and requires a slow (1s) transfer through a region of low field. Conversely here we require a very rapid transfer (faster than 100 ms) and substantially higher fields to *prevent* thermal mixing. The rapid-transfer is required also because the sample heats up very rapidly during the transfer, greatly boosting relaxation. We believe that a detailed analysis of (sample-specific) relaxation rates as a function of magnetic field and temperature is beyond the scope of this manuscript which intends to establish the general principle of the method. We have however conducted an experiment that, for the first time, reports on the sample heating rate during transfer (approximately 60 K/s) and include details of this measurement in the newly added Supplementary Material. We now cite the paper by Hirsch et al. at the beginning of the 4th paragraph and highlight the fact that Hirsch's approach is not compatible with DNP since DNP relies on radicals as a source of polarization. The positions of the two optical detectors are now stated explicitly in the manuscript.

Reviewer: (2) 'Scalable' in title. The audience except for specialists do not understand the meaning and significance of 'scalable' before read introduction. They may think that larger is usually more difficult. If the authors would like to use it, they should explain

why small sample is needed and difficult.

Response: The previously submitted manuscript states that rapid cooling in the cold polarizer leads to blockages if solvent amounts of less than 3 mL are used - that is why it is not possible to produce smaller amounts of hyperpolarized solutions with the conventional dissolution-DNP schemes. However, high-resolution NMR spectroscopy requires sample amounts between 100 μ L and 700 μ L. The use of small solvent amounts in high-resolution NMR is now emphasized in the abstract.

Reviewer: (3) Methods. Too technical for Nature Communications. The audience would like to pick-up key points of the work, which is prevented by the technical sentence which does not affect reproducibility. The authors should move the technical sentence to supplementary information. E.g. RS components and Python, etc. Also, the explanation of the instrumentation is difficult to understand. To number 1, 2, and 3 (or A, B, and C) for each important element and cite these in the sentence makes the explanation easier to understand.

Response: Substantial parts of the description of the instrument have been moved to the supporting information. Labels have been added to Figure 1 for increased clarity.

2 Reviewer 2

Reviewer: This paper describes development and demonstrations of dissolution NMR based on a game-changing idea. That is, samples in the solid-state with spin-hyperpolarized nuclei is shot from the region where nuclear hyperpolarization is performed into another place where enhanced NMR signal is measured. Unlike the previous strategy in which the frozen sample is rapidly dissolved with hot solvent before being transported to the NMR-measurement section, the authors firstly transfer the solid sample, then dissolve it, and finally measure NMR in the liquid state. To maintain the nuclear-spin polarization, the transfer is made as rapid as that of bullet-shooting, and in addition, the sample is led through a tunnel of a switchable magnetic field. The advantages of reversing the order of operations (i.e., transfer followed by dissolution instead of dissolution followed by transfer) are appropriately discussed, and preliminary experimental demonstrations are presented.

The idea behind this work is rather straightforward. Nevertheless, the activity of putting it into practice is very aggressive, and I find the work beautiful and praiseworthy. Even though what are already written in the manuscript are clear, detailed descriptions, in particular about the following important issues, are missing:

It is difficult for the reader to get Fig. 1(b)(c). The authors should indicate clearly the individual components that are important in Fig. 1(b)(c).

Response: Key individual components have now been labelled in the figure and are referenced accordingly in the figure caption.

Reviewer: Discussions on the minimum curvature of the tunnel for bullet-transfer are necessary.

Response: We have included a comment about the minimum curvature of the tunnel.

Reviewer: The Helmholtz coil for field reversal, which plays an important role, is explained only briefly in the main text, and it is very difficult to imagine how it works. A visual (photo and/or drawing) explanation is highly desirable.

Response: A sketch of the Helmholtz coil pair has been included in the newly added Supplemental Material and is referenced in the manuscript.

Reviewer: The asymmetric feature of the J-splitting resonance line (Fig. 3), which gives important information as to the attained spin polarization, would be obtained only when

the observing spin packets are selectively excited, or the tip angle of the excitation pulse is much smaller compared to 90 degree. However, no comment is found.

Response: The flip angle is indeed small and is now detailed in the caption of Figure 3, along with specific information about the chemical shifts of the main peaks. The manuscript has also been updated to include references that discuss the source of the smaller signals that were previously not assigned.

Reviewer: Overall, I believe the topic of the paper and the authors' achievement are impressive and suitable for publication in Nat. Comm., whereas the presentation needs revision. A separate supplementary material describing technical details would be helpful. **Response:** We thank the reviewer for his praise. Supplementary Material has been added to our manuscript.

REVIEWERS' COMMENTS:

Reviewer #1 (Remarks to the Author):

Authors correspond to my comment appropriately. In the revised manuscript, readers can understand the novelty and importance of their result. The manuscript describing the breakthrough of instrumentation should be published in Nature communications to open the new scientific field.

Reviewer #2 (Remarks to the Author):

In my review of the original manuscript, I was impressed by what is described but pointed out some issues on how the work is described. Now, after having read the revised manuscript, I recommend the paper for publication when the authors have considered the following issues further.

Discussions on the scalability raised by Reviewer #1 reminded me of the term "scalability" used in quantum computing, another field of growing general interest with many publications in Nature journals, where it is meant to "scale up", .i.e., increase the number of qubits. Conversely, it is interesting to see in the present context of Bullet DNP that "down-scalability" is the right direction. To make the work even convincing, I suggest seemingly minor but important (I believe) revision in the abstract,

From: scalable volume used in high-resolution...

To: scalable volume suitable for high-resolution...

The labeling in Fig. 1 in the revised version made the work much clearer, which, however, revealed yet unclear points:

- Sample transfer is described in detail, whereas the readers are unable to get the scheme of sample loading in the DNP part and sample evacuation in the NMR part. Regarding the former, the authors wrote that solution is "pipetted into a bullet", and then immersed in liquid nitrogen to freeze. Here, the "bullet" should be a hollow cylindrical container. In what follows, it is stated that the "bullet" is loaded into the DNP insert, and the frozen bullet of the sample is transferred. Now, if I am not mistaken, the "bullet" should be the bare, solid sample of frozen solution without the container. Did the authors broke the container to take the solid sample out? As for sample evacuation after NMR

measurement(s), the words "to facilitate evacuating" can be found in the caption of Fig. 1. However, it took me a while to figure out the way that the authors evacuate the liquid sample to enable further experiments without having to decompose the experimental setup. Another labeling in the Secondary Magnet in Fig.1a (red part just right to the pinch valve, I guess) would be helpful.

Kazuyuki Takeda

Response To Reviewer's Comments #2

1 Reviewer 1

Reviewer: Authors correspond to my comment appropriately. In the revised manuscript, readers can understand the novelty and importance of their result. The manuscript describing the breakthrough of instrumentation should be published in Nature communications to open the new scientific field.

Response: No further action is required.

2 Reviewer 2

Reviewer: In my review of the original manuscript, I was impressed by what is described but pointed out some issues on how the work is described. Now, after having read the revised manuscript, I recommend the paper for publication when the authors have considered the following issues further.

Discussions on the scalability raised by Reviewer #1 reminded me of the term "scalability" used in quantum computing, another field of growing general interest with many publications in Nature journals, where it is meant to "scale up", .i.e., increase the number of qubits. Conversely, it is interesting to see in the present context of Bullet DNP that "down-scalability" is the right direction. To make the work even convincing, I suggest seemingly minor but important (I believe) revision in the abstract, From: scalable volume used in high-resolution... To: scalable volume suitable for high-resolution...

Response: We agree and have changed the abstract accordingly.

Reviewer: The labeling in Fig. 1 in the revised version made the work much clearer, which, however, revealed yet unclear points:

- Sample transfer is described in detail, whereas the readers are unable to get the scheme of sample loading in the DNP part and sample evacuation in the NMR part. Regarding the former, the authors wrote that solution is "pipetted into a bullet", and then immersed in liquid nitrogen to freeze. Here, the "bullet" should be a hollow cylindrical container. In what follows, it is stated that the "bullet" is loaded into the DNP insert, and the frozen bullet of the sample is transferred. Now, if I am not mistaken, the "bullet" should be the bare, solid sample of frozen solution without the container. Did the authors broke the container to take the solid sample out? As for sample evacuation after NMR measurement(s), the words "to facilitate evacuating" can be found in the caption of Fig. 1. However, it took me a while to figure out the way that the authors evacuate the liquid sample to enable further experiments without having to decompose the experimental setup. Another labeling in the Secondary Magnet in Fig.1a (red part just right to the pinch valve, I guess) would be helpful.

Response: Throughout the manuscript, the term bullet refers to the PTFE sample container. We have now included a more detailed description of this container. Where possible we have clarified the wording further. The reviewer has assumed that the bullet is evacuated from the injection device - this is currently not implemented. Instead the NMR tube is evacuated in order to create a pressure gradient to drive the solution into the tube, as well as to prevent the formation of big air bubbles in the tube. We have now added a statement saying that the bullet remains in the injection device after the experiment, and that it has to be removed manually.